# Shedding a Light on the Challenges of Adolescents and Young Adults with Rhabdomyosarcoma

**DOI:** 10.3390/cancers14246060

**Published:** 2022-12-09

**Authors:** Andrea Ferrari, Susanne Andrea Gatz, Veronique Minard-Colin, Rita Alaggio, Shushan Hovsepyan, Daniel Orbach, Patrizia Gasparini, Anne-Sophie Defachelles, Michela Casanova, Giuseppe Maria Milano, Julia C. Chisholm, Meriel Jenney, Gianni Bisogno, Timothy Rogers, Henry C. Mandeville, Janet Shipley, Aisha B. Miah, Johannes H. M. Merks, Winette T. A. van der Graaf

**Affiliations:** 1Pediatric Oncology Unit, Fondazione IRCCS Istituto Nazionale dei Tumori, 20133 Milan, Italy; 2Cancer Research UK Clinical Trials Unit, Institute of Cancer and Genomic Sciences, University of Birmingham, Birmingham B15 2TT, UK; 3Department of Paediatric and Adolescent Oncology, Gustave-Roussy, Cancer Campus, Université Paris-Saclay, 94805 Villejuif, France; 4Pathology Department, Ospedale Pediatrico Bambino Gesù IRCCS, 00165 Roma, Italy; 5Pediatric Cancer and Blood Disorders Centre of Armenia, Yerevan 0014, Armenia; 6SIREDO Oncology Center, Institut Curie, PSL University, 75005 Paris, France; 7Tumor Genomics Unit, Department of Research, Fondazione IRCCS Istituto Nazionale dei Tumori, 20133 Milan, Italy; 8Department of Paediatric and AYA Oncology, Oscar Lambret Cancer Center, 59000 Lille, France; 9Hematology/Oncology, Ospedale Pediatrico Bambino Gesù IRCCS, 00165 Roma, Italy; 10Children and Young People’s Unit, Royal Marsden Hospital and The Institute of Cancer Research, Sutton SM2 5PT, UK; 11Department of Paediatric Oncology, Children′s Hospital for Wales, Heath Park, Cardiff CF14 4XW, UK; 12Hematology Oncology Division, Department of Women’s and Children’s Health, University of Padova, 35128 Padova, Italy; 13Department of Paediatric Surgery, University Hospitals Bristol and Weston NHS Foundation Trust, Bristol BS1 3NU, UK; 14Department of Radiotherapy, The Royal Marsden NHS Foundation Trust and The Institute of Cancer Research, Sutton SM2 5PT, UK; 15Sarcoma Molecular Pathology Team, Divisions of Molecular Pathology and Cancer Therapeutics, The Institute of Cancer Research, London SM2 5NG, UK; 16Sarcoma Unit, The Royal Marsden NHS Foundation Trust and The Institute of Cancer Research, London SM2 5NG, UK; 17Princess Máxima Center for Paediatric Oncology, 3584 CS Utrecht, The Netherlands; 18Department of Medical Oncology, The Netherlands Cancer Institute, 1066 CX Amsterdam, The Netherlands; 19Department of Medical Oncology, Erasmus MC Cancer Institute, Erasmus University Medical Center, 3062 PA Rotterdam, The Netherlands

**Keywords:** rhabdomyosarcoma, adolescents, young adults, AYA, review, clinical trial, age, outcome, treatment, biology, access to care

## Abstract

**Simple Summary:**

Rhabdomyosarcoma is the most frequent soft tissue sarcoma of childhood, but can occur at any age. Adolescent and young adult patients with rhabdomyosarcoma often have poorer outcomes than do children. This survival gap may be related to differences in clinical management or differences in tumor biology and intrinsic aggressiveness. Various studies have suggested that young adults may have better outcomes when treated with multidisciplinary treatment in line with the pediatric approach. However, treatment results seem to remain worse in young adults when compared with children, even when they were treated in the same way, and this suggest that part of the prognostic gap between children and adults may be attributable to biological differences in rhabdomyosarcoma arising in different age groups. A multifaced strategy is needed to further improve outcome of adults with rhabdomyosarcoma, including a trans-age academic societies and national/international cooperation, the definition of integrated biologic and genomic approach, and the development of collaborative rhabdomyosarcoma clinical trials without upper age limit.

**Abstract:**

Rhabdomyosarcoma (RMS) is a typical tumour of childhood but can occur at any age. Several studies have reported that adolescent and young adult (AYA) patients with RMS have poorer survival than do younger patients. This review discusses the specific challenges in AYA patients with pediatric-type RMS, exploring possible underlying factors which may influence different outcomes. Reasons for AYA survival gap are likely multifactorial, and might be related to differences in tumor biology and intrinsic aggressiveness, or differences in clinical management (that could include patient referral patterns, time to diagnosis, enrolment into clinical trials, the adequacy and intensity of treatment), as well as patient factors (including physiology and comorbidity that may influence treatment tolerability, drug pharmacokinetics and efficacy). However, improved survival has been reported in the most recent studies for AYA patients treated on pediatric RMS protocols. Different strategies may help to further improve outcome, such as supporting trans-age academic societies and national/international collaborations; developing specific clinical trials without upper age limit; defining integrated and comprehensive approach to AYA patients, including the genomic aspects; establishing multidisciplinary tumor boards with involvement of both pediatric and adult oncologists to discuss all pediatric-type RMS patients; developing dedicated projects with specific treatment recommendations and registry/database.

## 1. Introduction

Adolescents and young adults (AYA) with cancer are a unique set of patients characterized by specific features ranging from distinctive epidemiology and biology to clinical challenges, from multifaceted psycho-social issues to the necessity of holistic age-specific management [1,2]. These unique characteristics resemble neither those of children nor older adults with cancer, and are increasingly well recognised. Table 1 offers a summary of those challenging features that make AYA a special category of patients. While the definition of the AYA range is still under discussion, there is an emerging preference for the broader age range of 15–39 years, which highlights the transition of care between pediatric and adult oncologists.

In recent years, a variety of local, national and international programmes have been developed within different collaborative initiatives [3]. The increasing awareness within the European scientific community, for example, led the European Society of Paediatric Oncology (SIOPE) and the European Society for Medical Oncology (ESMO) to join forces, expertise and resources in the foundation of a dedicated AYA Working Group [4]. This group published a position paper to illustrate the vision shared by the two scientific societies and outlined the necessary steps to jointly deal with the most important issues. In particular, the inequitable access to care and clinical trials, and the survival gap reported in AYA patients for many neoplasms, including leukemias and lymphomas, brain tumors, and bone and soft tissue sarcomas [5].

Rhabdomyosarcoma (RMS) is one tumor type for which this survival gap is more evident. While it is a tumor typically occurring in the pediatric age range, RMS may in fact occur at any age, and poorer outcomes have been reported for AYA patients compared to children. A study from the North-American Surveillance, Epidemiology and End Results (SEER) database compared the clinical features and outcome of 1071 adults (age > 19 years) and 1529 children (age ≤ 19 years). The SEER study (period 1973–2005) demonstrated that adult patients were more likely to have adverse prognostic characteristics and far worse prognosis than children, i.e., 5-year overall survival: 26.6% versus 60.5% in children [6].

The epidemiological EUROCARE-5 study (study period 2000–2007) reported 5-year relative survivals of 66.6% in patients 0–14 years, 39.6% in those 15–19 years, and 36.4% in those 20–39 years [5].

Epidemiological studies also highlighted the different histological subtypes associated with age [6]. Pleomorphic RMS is completely different from the classic pediatric-type RMS (i.e., embryonal and alveolar subtypes), and occurs almost exclusively in the adult population. Pleomorphic RMS is currently seen as a pleomorphic sarcoma with myogenic rhabdomyoblastic differentiation, so a specific entity more similar (in biology, clinical behaviour and drug sensitivity) to other adult non-RMS high grade soft tissue sarcomas than to pediatric-type RMS [7,8,9,10].

This review discusses the specific challenges in AYA patients with pediatric-type RMS, exploring age and possible underlying factors which may influence different outcomes.

## 2. Rhabdomyosarcoma in Adolescent and Young Adult Patients: The Survival Gap

Though RMS is a highly malignant tumor, recent pediatric oncology studies report overall survival rates over 70% for patients with localized RMS treated with a risk-adapted multidisciplinary treatment approach including surgery, radiotherapy, and multi-agent chemotherapy [11,12,13]. The key elements contributing to progressive improvement in survival of RMS patients are; centralization of care to specialized centers, wide collaboration at national and international levels, and the high rate of inclusion in cooperative multi-institutional clinical trials [14].

In pediatric series, the outcome of RMS patients depends on multiple prognostic factors, such as histological subtype and molecular features, primary tumor anatomical site and size, lymph node involvement, and distant metastases at diagnosis [15]. Among these variables, age in itself has been identified as independently influencing survival [16,17]. As a consequence, pediatric RMS protocols include age as one of the prognostic factors and patients are stratified according to their age, with 10 years of age as a cut-off separating patients with a favourable prognosis (children less than 10 years) from those with a less good predicted outcome (older ones).

Whilst it is acknowledged that the survival of adolescents with RMS is inferior to that of children, outcome is reportedly even worse in adults, with published series showing survival rates in the 20–40% range [18,19,20,21,22,23,24,25,26,27,28]. Table 2 describes a selection of studies specifically focusing on adolescent and/or adult patients with RMS.

Reasons for such a survival gap are likely multifactorial. It might be related to differences in tumor biology and intrinsic aggressiveness, or differences in clinical management that could include patient referral patterns, time to diagnosis, enrolment into clinical trials, the adequacy and intensity of treatment, as well as patient factors including physiology and comorbidity that may influence treatment toxicity and tolerability, drug pharmacokinetics and efficacy [14,31,32,33].

## 3. Compliance with the Gold Standard Pediatric Therapeutic Guidelines

The concept that differences in outcomes might be related to differences in the administered treatment was supported by the historical study from the Istituto Nazionale Tumori in Milan on 171 adults (aged 18 years or older) who were treated between 1975 and 2001 [22]. This study confirmed the unfavourable clinical presentation of RMS in adults (50% of cases had the alveolar histotype, 60% had tumor arising in unfavorable sites, 32% had regional nodal involvement, 73% had tumors larger than 5 cm, all percentages being definitely higher than those generally observed in childhood), and the poor outcome (5-year event-free survival 27.9%, overall survival 39.6%). The study specifically analyzed the treatment received by each patient. Patients were stratified according to how closely their treatment approximated to current treatment guidelines for childhood RMS. A treatment score was assigned to each patient (Ferrari’s score).

Although all patients were treated in a national/international sarcoma-referral center, only 39% had received a treatment strategy similar to that used in childhood RMS. To obtain a score of 1 the patient required multimodal treatment with multidrug chemotherapy (including cyclophosphamide or ifosfamide, as well as anthracyclines and/or dactinomycin, for 8 cycles or more) and local treatment with surgery and/or radiotherapy. Using this scoring system, more than half of patients received potentially inadequate therapy: chemotherapy with different drugs than those used in pediatric RMS protocols, or chemotherapy given for much shorter duration (e.g., 3 cycles, as given in adult sarcomas), or chemotherapy not given at all (as done in some adult sarcomas after resection). A major finding of the report was that patients treated in line with the pediatric approach had clearly better outcomes (overall survival of 61% for patients with a score 1 versus 36% for those with a score < 1), suggesting that adult patients would fare better if they were treated with properly administered pediatric regimens.

Other studies also reported under treatment and discussed the poor compliance with pediatric RMS therapeutic guidelines as a cause of the survival gap observed in adults with RMS [24,25,26].

The possible barriers preventing adult patients from receiving optimal treatment remain a matter for discussion. Contributing factors may include: the lack of patient centralization to specialized centers; the absence of clinical trials, standardized therapeutic guidelines, or dedicated programs in adult organizations; the lack of experience of adult oncology teams to apply the key concepts of RMS therapy; the suboptimal collaboration between pediatric and adult sarcoma experts; and that adults may tolerate less well, treatments which have been designated for children [14].

AYA patients with RMS are consistently defined as a specific group outside the general adult population. This is underscored by the fact that even in the clinical practice guidelines for soft tissue sarcomas which are periodically updated by the European Society for Medical Oncology (ESMO) pediatric-types RMS are excluded. This emphasizes that these tumors should not be managed as other soft tissue sarcomas, but unfortunately increases the unfamiliarity with them among the adult sarcoma community [34].

Investigating the possible factors that prevent AYA patients with RMS from being treated optimally according to pediatric protocols is instrumental to identify areas for potential improvement, by addressing modifiable factors in the future.

In 2019, for example, a further study from the Milan group reported the outcome of a prospective series of 95 patients (aged 18–77) with embryonal and alveolar RMS treated between 2002 and 2015 [27]. This series presented the results achieved after the implementation of various measures aiming to improve the quality of treatment for adult patients with RMS, such as managing all adult RMS cases through a multidisciplinary discussion attended by both pediatric and adult oncologists, developing specific recommendations for the treatment of adult RMS (based on the principles adopted by pediatric protocols), and prospectively registering all adult RMS cases in a specific institutional database. The overall results of this series remained however unsatisfactory, with 5-year overall survival of 40.3%. This was influenced by the fact that 31% of the cases had metastatic disease at diagnosis (versus 17% in the previous study) [22]. When the treatment score was applied to the series, it was evident that the development of specific recommendations resulted in an improvement in the number of patients treated with intensive multimodal treatment resembling the pediatric strategy to 69.7% (versus 39% in the previous retrospective series). Patients treated in line with pediatric protocols had better outcome: in the group of adult patients with localized disease, overall survival was 58.8% in patients with score 1 and 30.3% in those with score < 1. This reinforced the idea that adherence to the principles of pediatric protocols, may improve adult RMS outcomes. However, the study demonstrated also that treating adults with a pediatric-type strategy was not enough to achieve results as good as those obtained in children. A further issue concerns treatment tolerance: 30% of patients did not receive the whole treatment compliant with pediatric principles, mainly due to chemotherapy-related toxicity (the scheduled chemotherapy was modified in 44% of the cases, while delays in the treatment administration were recorded in 57%) [27]. This may be related to difference in toxicity, but also to possible difference in medical attitude, with medical oncologists interacting with their adult patients that might be less rigorous in the translation of the pediatric guidelines into practice, accepting more modifications.

A further Italian study considered the therapeutic score to investigate to what extent the administered treatment was in line with the pediatric approach in a series of 104 RMS cases (excluding the pleomorphic subtype) diagnosed in Italy between 2000 and 2015. The study included 60 young patients (10–19 years old; mean age: 15 years old) and 44 adults (20–60 years old; mean age: 38 years old). A treatment score of 1 was assigned to 85% of younger patients, but only to 32% of older patients (*p* < 0.001). The significant impact of treatment on survival was confirmed in the multivariable analysis [29].

## 4. Access to Care

Pathway to diagnosis, referral to expert centers (and to facilities that have specific capacity for AYA patients), and inclusion into clinical trials are major issues for AYA patients presenting with RMS.

Several studies have described that adolescents often arrive at their tumor diagnosis after a significant delay. The detection of symptoms may be often neglected by adolescent patients, that refuse their parents’ attentions and mistrust adults generally, leading sometimes to a reluctance to see a doctor [35,36]. Longer symptom intervals may have an impact on the stage of the disease and final outcome [36]. This situation has also been specifically described for RMS patients [37].

It has also been widely reported that AYA patients receive less centralised care and limited enrolment onto clinical trials [38,39].

AYA patients suffer from decentralised care and a limited enrolment onto clinical trials [38,39].

With regards to sarcoma patients, various studies have demonstrated the key role of centralisation of care. Treatment at a high-volume centre and cooperation between pediatric and adult sarcoma experts within national joint networks have been shown to improve the quality of treatment and patient outcome [40,41,42,43]. AYA patients with RMS should be managed in experienced reference centres that participate in national and international sarcoma networks.

Referral to the sarcoma expert centre may ensure accurate histological diagnosis and staging investigations before embarking on the correct treatment protocol. Pathological diagnosis is particularly challenging due to the rarity of RMS. It is recommended that pathological diagnosis should be made by a pathologist with specific expertise in sarcomas, with the possibility to integrate histological diagnosis and molecular characterisation. The pediatric-type RMS subtypes include botryoid, embryonal, alveolar, and spindle cell/sclerosing RMS: with improved molecular characterisation, disease risk stratification is currentl considered based on PAX-FOXO1 (unfavourable) and MYOD-1 mutations (unfavourable) [10].

When the diagnosis is made outside of a reference centre or network, expert pathology validation is required.

Centralization of care should ensure a mandatory multidisciplinary approach: RMS can arise in any anatomical site so early communication with site specific surgical oncologists, medical and pediatric oncologists, and radiation oncologists is critical to plan tumor local therapy.

Adult RMS patients have historically not had access to pediatric RMS protocols, and cooperative prospective studies specifically dedicated to adult RMS have not been developed [14,44]. However, limited inclusion into clinical protocols has been observed also for adolescents, yet age cut-off criteria should not act as a barrier for eligibility. It is worth mentioning a report from the European pediatric Soft tissue sarcoma Study Group (EpSSG) which compared the number of patients enrolled in EpSSG clinical protocols with the number of cases expected to occur in the contributing European countries according to incidence rates. The analysis showed that adolescents were less represented in EpSSG protocols, even though the trials recruited patients up to 21 years of age; whilst 77% of the patients aged 0–14 years were included in EpSSG protocols, the percentage dropped to 64% for adolescents (15–19 years) [45].

An Italian study investigated where adolescents (15–19 years) with soft tissue sarcomas were treated in Italy, analyzing hospital discharge records obtained from the Health Ministry in the 2007–2014 period. Among the 381 cases, 239 (63%) were treated at 44 pediatric oncology centers part of the Italian Association of Pediatric Hematology and Oncology (Associazione Italiana Ematologia Oncologia Pediatrica AIEOP), while 142 (37%) were treated outside of the pediatric oncology national network, at non-AIEOP centers across 66 hospitals. Noteworthy, considering the patients treated at AIEOP centers, 55% of them were centralized in six high-volume centers. All the AIEOP centers enrolled patients in EpSSG protocols. Importantly, all the non-AIEOP centers were small-volume centers, and did not enroll patients in clinical trials [46].

## 5. The EpSSG Analysis

A major contribution to the discussion of access to care comes from the recent cohort study from the EpSSG, which analysed clinical findings, treatment data, toxicity and outcome of RMS patients registered onto the two clinical protocols, i.e., the EpSSG RMS 2005—phase 3 randomised trial for localised RMS (open from 2006 to 2016) [8,9]—and the EpSSG MTS 2008 protocol—prospective, observational, single-arm study for metastatic RMS (open from 2010 to 2016) [47]. The study aimed to compare AYA patients (defined here as those aged 15–21 years at diagnosis) with those aged 0–14 years old. The main purpose of the analysis was to ascertain whether the outcomes of AYA patients were persistently worse when compared to children, even when enrolled in the same clinical trials and receiving similar treatment (therefore eliminating the potential impact on survival of the lower recruitment into clinical trials and the possible undertreatment) [30]. The study involved 1977 patients (from 108 centres and 14 different countries), 1720 children and 257 AYA patients. Firstly, the analysis confirmed that AYA patients were more likely than children to have metastatic tumors (23.7% vs. 11.5%; *p* < 0.0001), unfavourable histological subtypes (46.3% vs. 26.2%; *p* < 0.0001), tumor > 5 cm (68.9% vs. 51.8%; *p* < 0.0001), and regional lymph node involvement (N1) (42.4% vs. 19.7%; *p* < 0.0001). Survival rates were significantly worse for AYA patients: 5-year event-free survival was 52.6% in AYA patients and 67.8% in children, 5-year overall survival was 57.1% and 77.9%, respectively (*p*-value < 0.0001). The multivariable Cox regression model confirmed the inferior prognosis of patient age ≥ 15 years. Survival rates remained significantly different according to age categories, when outcomes were analysed separately for different subgroups, with the exception of patients with localised embryonal RMS: in this subgroup, 5-year overall survival for AYA patients was 82.3%, similar to survival observed for children [30].

The EpSSG study demonstrated better results for AYA patients than those reported in epidemiological studies (as a rough comparison, 5-year survival was 57.1% for 15–21 year old patients treated between 2006 and 2016 in EpSSG protocols and 39.6% in 15–19 year old patients from EUROCARE-5 study which covered the 2000–2007 period) [5]. Together with other available studies, this finding supports the concept that AYA patients should be included in pediatric RMS trials to offer them the best chances to be cured. However, the EpSSG study showed that treatment results remained significantly worse in AYA patients when compared with children, even when they were treated in the same way. The only subgroup with similar outcome was that of patients with localised embryonal RMS [30]. All these findings would suggest that specifically tailored, but still intensive treatment strategy may be warranted for these patients.

## 6. Treatment Compliance and Tolerability

A further purpose of the EpSSG cohort study was to compare the treatment toxicity (and the modifications received as per protocol based on toxicity) in AYA patients and children. The inadequate compliance of adult patients to gold standard therapeutic guidelines has been ascribed to the inferior tolerability to intensive treatments originally developed for children (as reported, for example, in the study by Bergamaschi et al. already described) [27].

Differences in pharmacokinetics in relation to age—for example, in the metabolism of drugs usually utilised in RMS protocols such as vincristine, dactinomycin, and alkylating agents-has been described and considered potentially responsible for different treatment responses and toxicity [48]. In principle, a treatment protocol designed for children might be potentially less effective or too toxic for adult patients.

However, available data are contradictory. The North-American Intergroup Rhabdomyosarcoma Study (IRS)-IV showed that patients of 15–20 years of age experienced less hematological toxicity compared to younger patients receiving the same chemotherapy, while vincristine-associated neurotoxicity was higher) [49]. Similar results were reported from the Children Oncology Group (COG)-D9803 study [50]. On the other hand, the COG-ARST0431 study for metastatic RMS reported that adolescents (defined in that study as patients >13 years of age) were less likely than children to complete therapy (63% versus 76%) and more likely to have unplanned protocol dose modifications (23% versus 2.7%) [51]. Decreasing dose-intensity in vincristine, actinomycin-D, cyclophosphamide (VAC) chemotherapy for adult patients with RMS (patients aged greater than or equal to 21 years) was reported in a single-center retrospective study from Japan [52].

The EpSSG study did not report major toxicity or major protocol modifications in AYA patients compared with children. Grade 3–4 haematological toxicity and infection were observed more frequently in children than in AYA patients. Modifications of the chemotherapy programme were seen in 15% of patients aged 15–21 years and and 21% of patients younger than age 15 years [30]. These results, collected in a prospective multicentre study, are of great interest because they suggest that AYA patients, at least up to 21 years old, can be treated with intensive therapies originally designed for children, with no major tolerability issues. It remains to be clarified, of course, whether the same can be said for older adults.

## 7. Differences in Tumor Biology

Given that a survival gap remains even if AYA patients with RMS are treated on the same prospective protocols, part of the prognostic gap between children and adults may be attributable to biological differences in RMS arising in different age groups.

As a matter of fact, however, while our knowledge about the complex genomic landscape of pediatric RMS is constantly increasing [53,54,55], very few studies have addressed the age-related issues and there is still a shortage of information on the biology of RMS in adults. Understanding the tumor biology that differentiates childhood from adult RMS could provide critical information, open new therapeutic options and therefore drive new initiatives to improve patient outcome.

The underlying molecular differences in RMS subtypes according to age, are still subject to study. However, recent cohort studies of gentotype-phenotype characterization have contributed to a better definition of the subtypes and identification of new ones. In particular the spindle cell-sclerosing rhabdomyosarcoma (SCS-RMS), now recognized by the WHO classification for soft tissue tumors as a subgroup distinct from embryonal RMS and alveolar RMS, includes SCS-RMS with *MYOD1* mutation and SCS-RMS with *TFPC2* gene fusions [10,56,57,58] (Table 3).

SCS-RMS with *MYOD1* mutations are most frequent in AYA cases. MYOD1 mutations are almost exclusively a single mutation (c.365 T>G) in the MYOD1 sequence leading to the amino acid substitution MYOD1 L122R. The mutation is observed heterozygous or homozygous and is located in the conserved DNA binding domain of the MYOD1 transcription factor leading to transactivation and aberrantly taking over of MYC-like functions [59]. SCS-RMS with *MYOD1* mutation often occurs in AYA patients, especially in the head and neck region; a female predilection is generally observed in children, but in AYA there is a fairly equal distribution for males and females. Response to chemotherapy is reported lower than that usually observed in embryonal and alveolar RMS, and the outcome is worse, with high frequency of recurrence and death (reported in more than 50% of cases) [60]. The age-relationship of *MYOD1* mutation in RMS alongside with the poor outcome was very recently further confirmed in the largest cohort study to date with paired molecular profiling data and outcome data available. This study was a joint COG/UK retrospective analysis of 641 cases (age of patients ranged from 0 to 38 years, median 5.9): *MYOD1* mutations were identified in 17 of 515 fusion-negative cases (3%), and median age of these patients was 10.8 years (range 2–21 years). Interestingly, although SCS histology predominated, dense pattern of embryonal RMS and RMS not otherwise specified (NOS) were also found [61]. Latest global recommendations for molecular testing in RMS include the assessment of the *MYOD1* L122R mutation for risk stratification [62].

The *TFCP2* fusion positive subtype of SCS-RMS has initially been identified through RNA sequencing of sarcoma samples [58]. The function of the *TFCP2* fusion has not been completely clarified (e.g., aberrant transcription activity, interaction with RNA binding). RMS with *TFCP2* fusions do present mostly as intra-osseous disease, especially located in facial bones in young adults, but extra-osseous occurrence has been recently reported [63,64]. The limited data do not allow definite conclusions, however, the clinical behavior appears to be extremely aggressive [56,60]. Interestingly, the fusion partner of *TFCP2* in this entity is one of the FET proteins (i.e., *FUS* or *EWSR1*) a transcription factor and fusion partner well known from Ewing sarcoma and other sarcoma entities [65]. Whilst classified as RMS based on histological/immunohistochemistry appearance, the molecular driver may suggest a unique entity with different therapeutic needs. For example, the reported association of *TFCP2* fusion positive SCS-RMS with ALK overexpression might suggest a role for ALK inhibitors, although the potential benefits of ALK inhibition in this tumor type need to be further addressed [64].

With an increase in routine molecular profiling of RMS tumors and other diagnosis going forward, we will likely discover many more of these previously unknown molecular entities, with some of them possibly uniquely linked to the AYA and adult patient group.

A molecular group of alterations most commonly identified in fusion-negative RMS are *RAS* family mutations, including all three known isoforms, i.e., *HRAS*, *KRAS* and *NRAS* [61,66]. A correlation of the occurrence of *RAS* isoform mutations longitudinally with age has been demonstrated in the large cohort study including 515 fusion-negative RMS samples mentioned above [61]. Whilst there was an overall enrichment for all *RAS* isoforms in infants <1 year of age with a particular preponderance of *HRAS* mutations, adolescent patients showed a peak for *NRAS* mutations at the age of 13 years. The biological relevance of this is currently unclear and no correlation between individual RAS mutations and survival has been identified [61].

A genomic characterization study of the Memorial Sloan Kettering Cancer Center reported that somatic *TP53* pathogenic variants were more frequently detected in older patients: median age was 28 years in patients with TP53 mutations versus 14 years in TP53 wild type cases [67]. *TP53* pathogenic variants have recently been shown to be the key molecular prognostic marker (alongside *MYOD1* mutation) in the joint COG/UK retrospective analysis of 641 RMS tissue samples [61].

The Milan group recently published a study on microRNA (miRNA) and gene expression profiling (GEP) involving a cohort of 49 RMS cases, 28 children (0–14 years old) and 21 AYA cases (15–35 years) [68]. The study detected changes in the modulated immune contexture in AYA RMS, that could drive the aggressiveness of RMS in this particular age group. In detail, miRNA analysis revealed 39 miRNAs whose expression positively correlated with age, whereas 20 miRNAs were observed to decrease with age. Among all differentially expressed miRNAs, miR-223levels associated in pathway analyses with up-regulation of epithelial mesenchymal transition (EMT), immune cells function and inflammatory mechanisms. Gene expression analysis correlated with an up-regulation of several oncogenetic cancer mechanisms involved in the aggressiveness of AYA RMS. Interestingly, pathways associated with inflammation were observed as up-regulated in AYA cases, suggesting that both microenvironment and inflammation could play a pivotal role in RMS tumorigenesis of this age group. The only pathway that was down-regulated was myogenic differentiation suggesting that AYA RMS may be less differentiated compared to pediatric RMS. Finally, a cybersort analysis demonstrated an increase in genes associated with CD4 memory resting cells and a decrease in genes associated with T-cells in AYA RMS, confirmed also by IHC investigations [68]. Taking into accounts all these findings, the study suggested that AYA RMS aggressiveness might be explained by differences in microenvironmental signal modulation mediated by tumor cells, suggesting a fundamental role of immune context in AYA RMS development.

More recently, the same group investigated the possible predictive implication of chromosomal instabilities. Genomic Grade Index (GGI) and Complexity Index in Sarcoma (CINSARC) (a 67-gene prognostic signature related to chromosome integrity, mitotic control, and genome complexity in sarcomas) were analysed in pediatric and AYA cases. The study suggested a correlation of both signatures with RMS outcome, but no correlation with age. When tested independently on each age group, CINSARC correlated with outcome only in pediatric RMS, not in AYA patients [69].

Future translational research including comprehensive molecular profiling of frontline and relapsed RMS, across all ages, is likely to hugely enhance our understanding of how differences in biology between children and AYA with RMS may affect treatment response and survival. The currently recruiting protocol for RMS—the Frontline and Relapsed Rhabdomyosarcoma (Far-RMS) study (ClinicalTrials.gov Identifier: NCT04625907)—is open to RMS patients of all ages and provides an excellent platform and opportunity to facilitate this.

## 8. FaR-RMS: The Challenge of Integrating Adult Patients and Adult Centers into the Pediatric Network/Protocol

FaR-RMS is an overarching study for patients with newly diagnosed and relapsed RMS. Open in 2020, the study includes multi-arm, multi-stage randomised questions to evaluate: (i) systemic therapy through the introduction of new agent combinations in the most advanced disease stages (high/very high risk and relapsed RMS); (ii) the duration of maintenance therapy; and (iii) radiotherapy to improve local control for patients with higher risk of local failure (including patients aged 18 years or older), and to treat metastatic disease.

The FaR-RMS study has no upper age limit for recruitment, with the aim of evaluating whether the trial’s objectives lead to improved outcomes for RMS patients across the age spectrum. The molecular characterization of all cases included in the trial may help to better understand differences in disease biology in children and adults. The involvement of adult sarcoma experts in the FaR-RMS study should ensure AYA patients will be managed correctly. In addition, this should help understand efficacy, compliance and tolerability to treatment in older patients.

The FaR-RMS trial may represent a model of collaboration between pediatric and adult oncologists for the development of across all age disease specific protocols. Similarly, in United States current COG trials on RMS is adopting age limits for trial inclusion fixed at 50 years of age, to treat adults within the same protocols tailored for children and adolescents.

Cooperation is the key. Simply raising the upper age limits in pediatric RMS cooperative protocols to include adult cases, may only work if adult teams are included in the project (preferably from the beginning), otherwise adult physicians may not hear about such trials, or might be reluctant to enroll their patients in trials in which they themselves have no part. The success of the study regarding what concerns AYA patients, rest on the capability to establish a fruitful collaboration with the adult sarcoma community and the selection of the adult referral centers (in the different participating countries) to be involved. Addressing these challenges is a major goal for EpSSG in the coming years.

## 9. View for the Future and Conclusions

AYA patients with RMS are known to have worse outcome compared to children for a number of reasons, including the differences in disease biology and clinical presentation, the access to care, and the treatment administered [31,32,33,70]. However, improved survival has been reported in the most recent studies for AYA patients treated on pediatric protocols. Notably, the recent EpSSG study emphasized that AYA patients with fusion-negative RMS have similar outcome compared to younger patients and that AYA patients, at least up to 21 years old, can be treated with intensive therapies originally designed for children, with no major tolerability issues [30].

Different strategies may help to further improve outcome of AYA patients with RMS:(1)Support trans-age academic societies and collaborations at national and international level for patients with soft tissue sarcomas including RMS- the cooperation between the adult and pediatric sarcoma communities and the SIOPE-ESMO AYA Working Group may facilitate the development of specific initiatives and projects;(2)Develop specific clinical trials for pediatric-type RMS without upper age limit, such as FaR-RMS, to answer to clinical research questions and promote age-related biological and pre-clinical studies. Adult oncologists should be involved in the development of such trials from the beginning and direct patient involvement may be extremely helpful;(3)Define an integrated and comprehensive approach to AYA RMS, including the genomic aspects and associated translational research, including the development of molecularly fully characterised patient-derived xenografts and organoid cultures which will be analysed and studied in parallel with samples/cultures derived from pediatric RMS patients; this is essential to improve our knowledge of age-related biological factors and tumorigenesis and to potentially identify new targeted treatments. Innovative therapies (different from cytotoxic drugs) are warranted for metastatic patients (more frequently seen in older patients) and subtypes (i.e., SCS-RMS) that have shown to be less sensitive to standard therapy;(4)Establish multidisciplinary tumor boards (preferably national based given its rarity in adults) with formal involvement of both pediatric and adult oncologists to discuss all pediatric-type RMS patients;(5)Set up a national/international prospective registry/database for all adult/AYA cases with RMS, preferably alongside a clinical trial, and define specific treatment recommendations for adult patients that cannot be included, for different reasons, in cooperative protocols (such as FaR-RMS).

## Figures and Tables

**Table 1 cancers-14-06060-t001:** Specific challenges of adolescents and young adults (AYA) with cancer.

Issue	Uniqueness
Epidemiology	Unique epidemiology, with a wide range of cancer types, including those with a peak at pediatric and adult age
Biology and genetics	Many tumor types may have specific biology, that are different in AYA compared to children and older adults
	Age-specific molecular features are poorly understood for most AYA cancers
	Specific host biology, that differs according to age, with distinct pharmacokinetics and potential impact on therapy efficacy and toxicity
	Remarkable cases, related to cancer predisposition requiring genetic counselling
Awareness and pathway to diagnosis	Lack of awareness that cancer may occur in this age group, among general population as well as healthcare professionals
	Complex symptom appraisal process and pathway to diagnosis, risk of diagnostic delay, inappropriate access to specialized care
Clinical trial recruitment	Low recruitment rate in clinical trials compared to children
Survival	Lack of improvement in survival rates as compared to other age groups
	For some tumor types, survival in AYA is poorer than in children with the same disease
Reproductive function	Likelihood of infertility and potential reproductive problems create necessity for initial fertility preservation and age-specific counselling
Psychological and social care	Complicated psychological needs
	Complex communication challenges, shared decision-making, compliance and treatment adherence
	Necessity for age-specific psychological care, privacy issues, peer support
Palliative and end-of-life care	Challenging aspects in end-of-life care, related to the difficult adjustment to short life expectancy
Holistic approach	Need for a specific comprehensive multi-disciplinary team, involving professionals from various disciplines (e.g., psychologists, clinical nurses, social workers, youth workers, palliative care specialists, physiotherapists, occupational therapists, experts in fertility and sexuality)
	Involvement of both pediatric and adult medical oncologists
	Need for AYA special staff training and continuous professional education
	Need for age-appropriate clinical environments with dedicated facilities and programmes, tailored to their unique developmental needs
	Importance to give young people “voice and choice”; importance of partnership with patients advocates
Survivorship	Potential distinct clinical and psychological late effects, financial, educational and occupational challenges
	Transition into (older) adult medical system
	Need of promoting political and legal solutions to stop social discrimination and supporting the right to be forgotten

**Table 2 cancers-14-06060-t002:** Selected studies focusing on adolescent and/or adult patients with rhabdomysarcoma.

Series	Main Findings
Hawkins WG et al., 2001 [20] MSKCC, USA (1982–1999) Retrospective study	84 patients > 16 years (including pleomorphic cases); 5-year EFS 35% in all patients, 50% in patients < 20 years, <20% in older
Esnaola N. et al., 2001 [19] Boston, USA (1973–1996) Retrospective study	39 patients aged 16–82 years (median 27 years) (including pleomorphic cases); 5-year OS 31%
Little D et al., 2002 [21] MDACC, USA (1960–1998) Retrospective study	82 patients aged 17–84 years (median 27 years) (included pleomorphic); 10-year OS 40%
Ferrari et al., 2003 [22] INT Milan, Italy (1975–2001) Retrospective study	171 patients aged 18–83 years (median, 27 years) (including pleomorphic); 5-year EFS 28%, OS 40%; Patients stratified according to Ferrari’s treatment score; Only 39% of patients received treatment according to pediatric protocols (score 1); 5-year OS was 61.5% for score 1 and 36.5% for score < 1; Major conclusion: the use of pediatric protocols has an impact on prognosis
Joshi D. et al., 2004 [16] Pediatric North-American IRS Committee (1983–1997) Retrospective analysis of prospectively enrolled cases	2342 pediatric patients (<21 years); Adolescents (>10 years) had more unfavorable features and significantly poorer EFS than children aged 1–9 years (51% vs. 72%, *p* < 0.001); Major conclusion: age is an independent prognostic factor.
Sultan I. et al., 2009 [6] Epidemiological study: SEER database (1973–2005)	2600 patients, 1529 children (age ≤ 19 years) and 1071 adults (age > 19 years) (including pleomorphic cases); Adults had adverse prognostic variables and worse outcome (5-year OS 26.6% vs. 60.5%); Adults’ outcome remained significantly worse also analyzing subset of patients with similar tumors (i.e., same histotype, same stage, same sites)
Bisogno G, et al., 2012 [17] Pediatric Italian STSC (1988–2005) Retrospective analysis of prospectively enrolled cases	643 pediatric patients, 567 children (<14 years) and 76 adolescents (15–19 years); Only 27% of the expected adolescent patients were enrolled in Italian trials (vs. 90% of children); Adolescents had a longer symptom interval (8 weeks vs. 4.6 weeks); 5-year OS 68.9% in children vs 57.2% in adolescents;
Van Gaal C. et al., 2012 [23] Multicenter Dutch study (1977–2009) Retrospective study	169 patients all ages, 118 children (<16 years) and 51 adults (≥16 years); 5-year OS 64.8% in children, 21.4% in adults; Age was an independent prognostic factor
Dumont S. et al., 2013 [24] MDACC, USA (1957–2003) Retrospective study	239 patients >10 years, 122 <20 years, 117 ≥20 years (including pleomorphic cases); Age > 50 was significantly predictor of worse outcome; Multimodality therapy was significantly associated with longer OS
Gerber NK. et al., 2013 [25] MSKCC, USA (1990–2011) Retrospective study	148 patients > 16 years (median age 28 years) (including pleomorphic); 5-year OS was 54% for protocol patients vs. 36% for non-protocol patients;
Fischer D. et al., 2018 [26] Epidemiological study: National Cancer Database (1998–2012)	2312 patients, 1021 aged < 15 years, 507 AYA (15–39 years) and 784 older adults (age ≥ 40 years) (including pleomorphic cases); Adults received multimodal therapy least often, i.e., pediatric: 62%, AYA: 46%, adults: 24%; Multimodal therapy was associated with a decreased risk of death;
Bergamaschi L. et al., 2019 [27] INT Milan, Italy (2002–2015) Prospective study	95 patients (age 18–77 years, median 27) with pediatric-type RMS (pleomorphic histotypes excluded); 5-year EFS 33.6%, OS 40.3%; Treatment score: in localized disease, 5-year OS 58.8% for score 1 and 30.3% for score < 1; chemotherapy-related toxicity caused treatment modifications and delays in many cases;
Drabbe C. et al., 2020 [28] RMH, UK (1990–2016) Retrospective study	66 patients (age 18–71, median a 28) with pediatric-type RMS (pleomorphic histotypes excluded); 5-year OS 27%; localized tumor: 5-year OS 36%; metastatic tumor: 5-year OS 11%
Ferrari et al., 2021 [29] Italian cancer registries (2000–2015); Retrospective study	104 patients, 60 aged 10–19 years, 44 aged 20–60 years old, with pediatric-type RMS (pleomorphic histoypes excluded); Treatment score: score of 1 assigned to 85% of 10–19 year-olds and 32% of >20 years; treatment score was an independent prognostic factor at multivariable analysis
Ferrari et al., 2022 [30] EpSSG (2005–2016) cohort study of prospective protocols (EpSSG RMS 2005 and MTS 2008)	1977 patients, 1720 children (0–14 years) and 257 AYA (15–21 years) (pleomorphic histotypes excluded); 5-year OS was 57.1% and 77.9% in AYA and children (*p*-value < 0.0001): survival worse in AYA than in children, even when they were treated in the same way; No major toxicity or major protocol modifications in AYA compared with children: AYA patients up to 21 years old, can be treated with intensive therapies originally designed for children

MSKCC = Memorial Sloan-Kettering Cancer Center, New York, USA; MDACC = MD Anderson Cancer Center, Houston, USA; INT = Istituto Nazionale dei Tumori, Milan, Italy; IRS = Intergroup Rhabdomyosarcoma Study Committee; SEER = Surveillance, Epidemiology and End Results; STSC = Soft Tissue Sarcoma Committee; RMH = Royal Marsden Hospital; EpSSG = European pediatric Soft tissue sarcoma Study Group; AYA = adolescents and young adults; EFS = event-free survival; OS = overall survival.

**Table 3 cancers-14-06060-t003:** Rhabdomyosarcoma (RMS) subtypes according to age.

RMS Subtypes According to the WHO Classification		Findings According to Age
1	Embryonal RMS	Better prognosis; it is the most frequent RMS in children
2	Alveolar RMS with FOXO1 fusions	PAX3-FOXO1 and PAX7-FOXO1! Worse prognosis; it accounts for 25% of RMS in children and 50–60% of RMS in adults
3	Spindle cell-sclerosing (SCS) RMS	Three molecular subgroups: (a) infantile SCS RMS with VGLL2::NCOA2 rearrangements—good prognosis; exclusive of young children (b) SCS RMS with MYOD1 mutations—most frequent in AYA cases, especially in head/neck region; lower response to chemotherapy, worse outcome (c) SCS RMS with *TFPC2* gene fusions—most frequent in adults; aggressive clinical behaviour
4	Pleomorphic RMS	Specific entity occurring almost exclusively in the adult population; more similar to other adult non-RMS high grade soft tissue sarcomas than to pediatric-type RMS; lower response chemotherapy, aggressive behaviour

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
