# Peer review of "Shedding a Light on the Challenges of Adolescents and Young Adults with Rhabdomyosarcoma"

_cancers, 2022, doi:10.3390/cancers14246060_

Round 1

Reviewer 1 Report

This extensive and well-written review aims to discuss the specific challenges in AYA patients with pediatric-type RMS, exploring possible underlying factors which may influence different outcomes. If restricted to embryonal-& alveolar type(s) in AYAs (excluding the more common pleomorphic subtype in older patients), there are, in fact 2 types here!

It presents just the “EUROPEAN PERSPECTIVE with corresponding author –string – how does it compare to the US perspective? Could be mentioned in the DISCUSSION.

Table 2 versus TABLE 1: Several items; like Definition..Epidemiology and more are listed both places. Tables contain to a large degree the same info / …..adjust/avoid overlap…Merge into one Table?

Strictly excluding pleomorphic-subtype will, most likely explain most of the poor outcomes reported in the several publications cited  (probably  more than some of the other factors listed by the authors?)

I would like to challenge the authors to condense (may be also shorten the text) and structure the manuscript somewhat better. I would have liked to see a more hierarchal disposition. Starting with comparative histologies, impacts from tumorbiology…and then later, topic by topic, to other factors like access to treatment/dose-intensity/compliance to treatment and, at the end, social/psychological factors.

From reading this review, it seems to me, not clear that, when comparing survival figures in the several publications cited, it may be  comparing apple & oranges….between the three (even more molecularly defined entities) histological sub-types --- RMS “different diseases” between age-groups, different distributions of disease stages (size, nodal involvement and/or metastases)  at primary diagnosis. These may influence/contribute more to dismal outcomes in AYAs (& adults) than the actual given treatment?

MINOR: Avoid abbreviation in the section HEADINGS

Author Response

Reviewer 1

Comment: It presents just the “EUROPEAN PERSPECTIVE with corresponding author –string – how does it compare to the US perspective? Could be mentioned in the DISCUSSION.

Response: We thank the Reviewer for the comment. This is true. The paper is part of the Cancer Special Issue “Improving treatment for Rhabdomyosarcoma: a collaborative approach through the European paediatric Soft tissue sarcoma Study Group (EpSSG)”. The manuscript describes the European perspective, but it is extremely important to compare it to the US experience. Data on COG studies regarding AYA are described in the text. In the revised version, at page 21 we also added some lines to discuss as, in US, current COG trials on RMS is adopting age limits for trial inclusion fixed at 50 years of age, to treat adults within the same protocols tailored for children and adolescents. 

Comment: Table 2 versus TABLE 1: Several items; like Definition..Epidemiology and more are listed both places. Tables contain to a large degree the same info / …..adjust/avoid overlap…Merge into one Table?

Response: Table 1 and table 2 are very different. Table 1 briefly presents the specific challenges of AYA with cancer. It does not refer to rhabdomyosarcoma patients, but more in general to young people with cancer: it describes the unique characteristics that make AYA a special category of patients. We believe that this table is important as introduction, to set the scene on the challenges of AYA patients. In the revised version, we have shortened the table and we have eliminated repetitions (for example, the definition of AYA is already present in the main text and has been removed from the table).

Table 2 focuses on AYA with RMS and present the main published studies on this patient category, describing the series (number of patients, study period, histotypes included with a specification on pleomorphic RMS) and the main features/conclusions of any study. We believe that this table has a central role in the paper and offer to the readers an important tool, with the review of main literature. However, we have revised also table 2 to shorten and simplify it. We hope that this may considered acceptable by the Reviewer.

Comment: Strictly excluding pleomorphic-subtype will, most likely explain most of the poor outcomes reported in the several publications cited  (probably more than some of the other factors listed by the authors?)

Response: Pleomorphic RMS certainly account for part of the survival gap observed in adult RMS in historical series. However, survival gap remains also when only embryonal and alveolar subtypes are considered (and pleomorphic cases are excluded). Table 2 clearly specifies if pleomorphic cases are included or excluded for each series.  In particular, pleomorphic RMS cases were excluded in the most recent studies discussed in the text, i.e. the INT Milan series by Bergamaschi et al (2019), the UK study by Drabbe et al (2020), the Italian study by Ferrari et al (2021) and the EpSSG study.

Comment: I would like to challenge the authors to condense (may be also shorten the text) and structure the manuscript somewhat better. I would have liked to see a more hierarchal disposition. Starting with comparative histologies, impacts from tumorbiology…and then later, topic by topic, to other factors like access to treatment/dose-intensity/compliance to treatment and, at the end, social/psychological factors.

From reading this review, it seems to me, not clear that, when comparing survival figures in the several publications cited, it may be  comparing apple & oranges….between the three (even more molecularly defined entities) histological sub-types --- RMS “different diseases” between age-groups, different distributions of disease stages (size, nodal involvement and/or metastases)  at primary diagnosis. These may influence/contribute more to dismal outcomes in AYAs (& adults) than the actual given treatment?

Response: We have tried to short the text a little bit, avoiding repetitions. Concerning the structure and the disposition, we may agree that starting from histology and biology, and then prognostic factors and then treatment can be a more classic structure. However, the manuscript was not thought as a book chapter, or as a classic review paper. We wanted to shed a light on the problem of survival gap between AYA and children with RMS, discussing the possible reasons, the challenges and the solutions. For this reason, we believe it could be more interesting to start from the survival differences. This approach is exactly what happened when clinicians started to look at the survival gap, and then tried to explore the possible differences in treatment approaches, and then only recently investigated possible differences in tumor biology.

We believe that the discussion may flow potentially better, in this way. We hope that the Reviewer might accept this approach.

Concerning the problem to compare different studies, this is true. But it is what happen always when someone tries to compare outcome and treatment of patients with heterogeneous diseases, treated in different places and in differents ways. RMS is a very heterogeneous tumor… apples and oranges, or maybe Gala, Golden Delicious, and Granny Smiths apples. We have tried to better clarify in text and tables what studies included pleomorphic RMS and what only pediatric-type RMS, for example; or the age range of involved patients. The revised table 2 should be help the readers to better see what we are talking about.

Comment: Avoid abbreviation in the section HEADINGS

Response: We have modified the text, avoiding abbreviation as RMS and AYA in Headings, as suggested.

Reviewer 2 Report

The authors provide a review of the current state of treatment for adolescents and young adults (AYAs) with rhabdomyosarcoma and the challenges to improving their rates of survival. The authors highlight key factors that contribute to adverse outcomes including unique tumor biology and aggressiveness, heterogeneous treatment patterns depending on the location of care, and tolerability of treatment due to toxicity in older populations. The authors cite evidence that treatment on pediatric protocols can improve outcomes in AYA RMS and provide support for integrated adult and pediatric collaborative approaches to their care. 

Overall the manuscript is well written though it would benefit from review from an editor to ensure the English language and style are suitable. I have a couple of specific comments:

1) Table 2 in the manuscript available for review does not show studies focusing on AYA RMS, but is a copy of Table 1.
2) Under section 4, two consecutive sentences are redundant and one should be removed: either 1) "It has also been widely reported that AYA patients receive less centralised care and limited enrolment onto clinical trials", or 2) "AYA patients suffer from decentralised care and a limited enrolment onto clinical trials".

Author Response

Comment: Table 2 in the manuscript available for review does not show studies focusing on AYA RMS, but is a copy of Table 1.

Response: There was probably a mistake in the submission. Table 1 and table 2 are different tables. In the current version, we hope the two tables are clearly represented. Table 1 briefly presents the specific challenges of AYA with cancer. Table 2 focuses on AYA with RMS and present the main published studies on this patient category, describing the series (number of patients, study period, histotypes included with a specification on pleomorphic RMS) and the main features/conclusions of any study.

Comment: Under section 4, two consecutive sentences are redundant and one should be removed: either 1) "It has also been widely reported that AYA patients receive less centralised care and limited enrolment onto clinical trials", or 2) "AYA patients suffer from decentralised care and a limited enrolment onto clinical trials".

Response: We have modified the text accordingly.

Reviewer 3 Report

The review entitled “Shedding a light on the challenges of adolescents and young adults with rhabdomyosarcoma,” provides a wonderful insight into the differences evident between AYA vs pediatric and adult RMS. It provides many clinically-relevant perspectives that are critically needed to help guide treatment. Here are a few suggestions to improve and make the review a bit more comprehensive.

·         Table 1 and Table 2 are essentially the same- can they be written differently or combined?

·         Page 3: European Society of Paediatric Oncology (SIOPE) should be SIOP not SIOPE

·         Table on different types of RMS will be good to include

·         On page 9 the authors state, “SCS-RMS with MYOD1 mutation are most frequent in AYA cases,” is the MYOD1 mutation increases or decrease MYOD1 function?

·         Does SCS-RMS with MYOD1 mutation occur more frequently in AYA patients with certain racial/ethnic backgrounds or in certain genders?

·         Similarly, for the TFCP2 fusion positive SCS-RMS, does it occur more frequently in AYA patients with certain racial/ethnic backgrounds or in certain genders?

·         Does the TFCP2 fusion protein act as an aberrant transcription factor?

·         A table comparing adult RMS vs pediatric and AYA RMS would be good for visualization

·         PAX3-FKHR1 or PAX7-FKHR fusion RMS are not really discussed in the review. Do these fusions occur in AYA or adult patients? It would be good to include a little bit on these types of fusion-positive and fusion negative RMS patients.

·         A figure on pathways activated in different types of RMS would be good to include so that readers can see all the different pathways that could be activated.

·         The review is great in terms of covering the different clinical trials but a section on what is being done at the preclinical level would also help make this review more comprehensive.

·         In the future directions section it may be good for authors should discuss use of targeted therapies and/or concept of precision medicine in rhabdomyosarcoma.

·         Current front-line and second-line therapies for RMS patients can have toxic effects on quality of life. The authors mention fertility issues but what other things can occur? Can this be discussed a bit more?

Author Response

Comment: The review entitled “Shedding a light on the challenges of adolescents and young adults with rhabdomyosarcoma,” provides a wonderful insight into the differences evident between AYA vs pediatric and adult RMS. It provides many clinically-relevant perspectives that are critically needed to help guide treatment.

Response: We thank the Reviewer for the nice words

Comment:  Table 1 and Table 2 are essentially the same- can they be written differently or combined?

Response: Table 1 and table 2 are very different. Table 1 briefly presents the specific challenges of AYA with cancer. It does not refer to rhabdomyosarcoma patients, but more in general to young people with cancer: it describes the unique characteristics that make AYA a special category of patients. We believe that this table is important as introduction, to set the scene on the challenges of AYA patients. In the revised version, we have shortened the table and we have eliminated repetitions (for example, the definition of AYA is already present in the main text and has been removed from the table).

Table 2 focuses on AYA with RMS and present the main published studies on this patient category, describing the series (number of patients, study period, histotypes included with a specification on pleomorphic RMS) and the main features/conclusions of any study. We believe that this table has a central role in the paper and offer to the readers an important tool, with the review of main literature. However, we have revised also table 2 to shorten and simplify it. We hope that this may considered acceptable by the Reviewer.

Comment: Page 3: European Society of Paediatric Oncology (SIOPE) should be SIOP not SIOPE

Response: We suppose there may be a small misunderstanding here: SIOP is the acronym for the International Society of Pediatric Oncology, while SIOPE is for the European Society of Paediatric Oncology. In the text, we refer to SIOPE and the creation of an European Working Group dedicated to AYA, together with the adult European society ESMO (European Society for Medical Oncology)

Comment: Table on different types of RMS will be good to include

Response: Following the suggestion, we have added a new table (Table 3) which describes RMS subtypes according to age.

Comment: On page 9 the authors state, “SCS-RMS with MYOD1 mutation are most frequent in AYA cases,” is the MYOD1 mutation increases or decrease MYOD1 function?

Response: To clarify this point, we have added the following sentences at page 16: “MYOD1 mutations are almost exclusively a single mutation (c.365 T>G) in the MYOD1 sequence leading to the amino acid substitution MYOD1 L122R. The mutation is observed heterozygous or homozygous and is located in the conserved DNA binding domain of the MYOD1 transcription factor leading to transactivation and aberrantly taking over of MYC-like functions.” A more in deep discussion on MYOD1 will be present in other papers of the Cancers Special Issue dedicated to RMS, that includes also the current manuscript. The special issue is entitled “Improving treatment for Rhabdomyosarcoma: a collaborative approach through the European paediatric Soft tissue sarcoma Study Group (EpSSG)”.

Comment: Does SCS-RMS with MYOD1 mutation occur more frequently in AYA patients with certain racial/ethnic backgrounds or in certain genders?

SCS-RMS with MYOD1 mutation are most frequent in AYA cases, especially head and neck cases; Response: We have specified in the text that “a female predilection is generally observed in children, but in AYA there is a fairly equal distribution for males and females”. No ethnic predilection has been described.

Comment: Similarly, for the TFCP2 fusion positive SCS-RMS, does it occur more frequently in AYA patients with certain racial/ethnic backgrounds or in certain genders?

Response: No clear gender or racial predilection is described.

Comment: Does the TFCP2 fusion protein act as an aberrant transcription factor?

Response: This is an important question. Insufficient research has been done into this question to clearly state the function of this fusion. Whilst it is very likely that aberrant transcription activity plays a role in the disease – and we tried to point to this by describing the aberrant ALK expression which has been found - other functions of the fusion protein such as interaction with RNA binding through the fusion partner EWSR1 could play a key role. We have added a sentence to briefly clarify this point at page 17. Other papers of the Cancers Special Issue will focus on RMS biology and better focus on these aspects. 

Comment: A table comparing adult RMS vs pediatric and AYA RMS would be good for visualization.

Response: We have added a new table (Table 3) which describes RMS subtypes, with specific highlights according to age. We hope that this may be good for visualization.  

Comment: PAX3-FKHR1 or PAX7-FKHR fusion RMS are not really discussed in the review. Do these fusions occur in AYA or adult patients? It would be good to include a little bit on these types of fusion-positive and fusion negative RMS patients.

Response: The distribution of the different histological types in AYA patients vs children, has been discussed in the text. For example, we highlighted at page 13 that in the EpSSG series unfavourable histological subtypes (that mean alveolar RMS in such series on pediatric-type RMS) was 46.3% in AYA vs 26.2% in children; p<0.0001.

In the new added table, we have specified the different proportion of embryonal/alveolar RMS according to age.

As outline in the review, only limited molecular analysis has been performed in adult patients with RMS so far to further discuss this, for example for the presence of PAX3-FKHR1 or PAX7-FKHR fusion.

As comprehensive biology is planned alongside the ongoing FaR-RMS study which includes all ages, we are hoping to also close this gap and be able to learn about distribution and potential clinical/translational meaning of different genomic alterations in different age groups including the common fusions (PAX3-FOXO1 and PAX7-FOXO1) in alveolar RMS.

Comment: A figure on pathways activated in different types of RMS would be good to include so that readers can see all the different pathways that could be activated.

Response: As our manuscript focuses on the overall differences and special needs of AYA patients with RMS rather than the biology of this tumor we would like to kindly refer the Reviewer to other reviews in this Special Issue which will focus on biology and are likely to provide such pathway overview. The principles of pathway activation and targetability and the challenge to target the aberrant transcription factor fusion genes is not genuinely different in pediatric and young adult/ adult RMS. As outlined, whilst differences exist between age groups, the extent and molecular nature is insufficiently understood as not enough samples from young adults have been profiled.

Comment: The review is great in terms of covering the different clinical trials but a section on what is being done at the preclinical level would also help make this review more comprehensive.

Response: We thank the Reviewer for the comment. As also outlined above, the knowledge about molecular alterations in AYA RMS is scarce and all activities/recent publications we are aware of have been summarized in the session dedicated to possible differences in biology according to age. A general review of preclinical studies in RMS above the aim of our paper, and will be covered in other papers of the Special Issue.

What is needed is exactly what you are asking for, i.e. more research on the preclinical and translational level. We highlighted this in our review, where we describe the currently ongoing FaR-RMS study which is covering all ages. In addition to emphasize the importance of more research, we have added a few lines with regards to translational research in point 3 of the “View for the future and conclusions“ section, talking about “translational research including the development of molecularly fully characterised patient-derived xenografts and organoid cultures which will be analysed and studied in parallel with samples/cultures derived from pediatric RMS patients”.

Comment: In the future directions section it may be good for authors should discuss use of targeted therapies and/or concept of precision medicine in rhabdomyosarcoma.

Response: We thank the Reviewer for the comment. As outlined biological and molecular information on RMS in AYA is scarce, hence we do not yet have this information to discuss targeted therapies and precision medicine for this age group. We did however mention for example the ALK overexpression in most patients with TFCP2 fusions, but that the benefit of ALK inhibitors in this patient group is currently unclear. In addition, in the “View for the future and conclusions“, we underline that ”innovative therapies (different from cytotoxic drugs) are warranted for metastatic patients (more frequently seen in older patients) and subtypes (i.e. SCS-RMS) that have shown to be less sensitive to standard therapy”. Again, there will be other review papers which are likely to discuss this in the Special Issue. We thank in advance for understanding.

Comment: Current front-line and second-line therapies for RMS patients can have toxic effects on quality of life. The authors mention fertility issues but what other things can occur? Can this be discussed a bit more?

Response: Toxic effects are of great importance, in particular the issue of late sequelae. Since the paper focus on the comparison of RMS in AYA and RMS in children, we discussed the problem of treatment tolerance according to age. In particular, the EpSSG cohort study compared the treatment toxicity in AYA and children (given that other studies suggested inadequate compliance of adult patients to gold standard therapeutic guidelines, as results of inferior tolerability to intensive treatments originally developed for children). Aldo data from COG studies are reported.

Concerning the issue of late sequelae, instead, we believe that a dedicated discussion may be above the scope of a manuscript aiming to compare adult and children with RMS. We hope that this may the considered acceptable be the Reviewer.